# Research Progress on the Influence of Thermo-Chemical Effects on the Swelling Pressure of Bentonite

## Jinjin Liu, Chuanqin Yao *, Wenbo Su and Yizhe Zhao

School of Civil Engineering, Shanghai Normal University, Shanghai 201418, China; yy21_anhuibbu@163.com (J.L.)
* Correspondence: cqyao@shnu.edu.cn

**Abstract:** The swelling pressure of bentonite changes dramatically due to diffused nuclear radiation heat and underground osmosis, causing the failure of the buffer isolation layer in deep geological repositories for the disposal of high-level radioactive waste. A detailed overview of the relevant research results on the swelling pressure variation of bentonite under thermo-chemical effects is presented in this paper. The results showed that the values of the swelling pressure obtained by different test methods are dissimilar. The swelling pressure of bentonite decreased with the increasing pore solution concentration; nevertheless, the effect of temperature on the swelling pressure is still controversial. At the micro-level, crystal layer swelling and double-layer swelling are generally considered to be the main factors affecting the swelling pressure; the pore structure and water distribution of bentonite will change owing to thermo-chemical effects. At the macro-level, involving intergranular stress, a mechanical parameter was proposed to explain the mechanism of the changes in the swelling pressure of bentonite. Finally, future research directions for the study of the evolution of bentonite swelling properties under thermo-chemical effects are proposed, based on the current research results.

**Keywords:** thermo-chemical; swelling pressure; test methods; microscopic properties; intergranular stress





## 1. Introduction

With the rapid evolution of nuclear energy, a large amount of high-level radioactive waste (HLW) has been produced [1]. Disposing of this high-level radioactive waste safely, in an environmentally friendly way, and effectively for a long time has become a major problem that requires urgent solutions worldwide. At present, the most effective and feasible method of disposal of HLW is deep geological disposal, which requires choosing a suitable site (crust with poor water content, good stability, far from human activity zones, etc.) to set up an engineering barrier, buffering and sealing the tanks containing HLW in a deep geological repository at a depth of 500–1000 m from the surface to prevent the waste from being transported to the biosphere in case of a spill [2]. Bentonite, which contains a large amount of montmorillonite, is internationally recognized as a buffer/backfill material for the deep geological disposal of HLW due to its characteristics of low permeability, large swelling, strong adsorption, and good thermal conductivity [3,4]. Furthermore, bentonite not only has the important characteristics of buffering and backfilling, but also has the advantages of low cost, wide distribution, easy mining, and easy processing; so, it has become the preferred buffering backfilling material around the world. In China, Gaomiaozi bentonite produced in Xinghe County, Inner Mongolia, was selected preliminarily as the base material for buffering/backfill material [5,6].

Bentonite is in an unsaturated state after the construction of deep geological repositories. As the groundwater level rises, the compacted bentonite absorbs water (solution) undergoing an expansive deformation that can tightly enclose the waste tank and fill the gap between the buffer material and the surrounding rock mass, fully blocking the surrounding

rock and the waste tank. In the meantime, a sufficiently large swelling pressure under almost constant volume conditions can improve the sealing efficiency of bentonite [7–9], which is the main governing factor for the stability of subsurface structures. However, the near-field chemical conditions are complex during the operation of an HLW repository [10], as bentonite draws water from the surrounding rock and becomes gradually saturated due to subsurface infiltration. This will result in changes in the chemical concentration of bentonite pore water. Moreover, the temperature in the repository will be changed on account of the diffuse nuclear radiation heat. In the long term, problems such as the loss of bentonite's swelling properties may arise. Therefore, the study of bentonite's swelling performance under coupled chemical and temperature gradients field conditions is a crucial issue for the safe operation of nuclear waste repositories.

The swelling characteristics of compacted bentonite have been extensively investigated by scholars in the past decades. Focusing on this theme, this paper systematically describes the progress and limitations of the studies on bentonite swelling pressure under the effect of thermochemistry, with regard to the experimental test methods, the law and mechanism of swelling pressure variations under the influence of pore solution concentration and temperature, etc. On this basis, an outlook of future research developments to understand the influence of thermo-chemical effects on bentonite swelling pressure is provided.

## 2. Test Methods for Swelling Pressure

The swelling pressure can be measured by different methods, but different test methods produce different swelling pressure values [9,11,12]. It should be emphasized that it is ideal to think of the swelling pressure as the pressure measured when bentonite is wetted under the condition of limited volume [13]. The constant volume method, constant load method, and paired swell test (PS-test) were proposed to measure the swelling pressure [14].

The constant volume method for measuring the swelling pressure is based on strain control techniques [15,16] including the zero-swell test and the direct method. The main difference between the two approaches is that the latter measures the swelling pressure directly using a pressure transducer, such as the S-type load cell shown in Figure 1. The zero-swell method prevents the specimen from expanding by applying a vertical stress, which is generally around the year 2000 [11,17]. It is challenging to guarantee optimal swelling readings and constant volume conditions during the test, and the soil pressure is very sensitive to load increments during the test [9,18], especially in the zero-swell test [14]. Furthermore, the swelling pressure will be overestimated due to the friction between the specimen and the consolidator ring during the recovery of the specimen to the initial volume [9]. In spite of this, the value measured with the zero-swell method is only one-third of the value measured with the swelling reloading test [9,17]. The reason is that the soil particles in the swelling reloading test are sufficiently hydrated, especially under low load conditions, whereas soil particle swelling is restricted and hydration is not sufficient in the constant volume method. Large errors with the direct method are due to the device's characteristics, including the stiffness of the sensor [19,20], and the corresponding swelling pressure error can reach a maximum of 1–2 MPa [9].

The constant load test methods include the swell-consolidation test and the double consolidation test. The swell-consolidation method involves compressing the specimen after free swelling for 24 h until it returns to its initial state of porosity or height, which is typical of the constant load method [14]. The swelling pressure is the pressure necessary for the sample to return to its initial height [11], similarly to the swelling pressure of the free-swelling method. The difference between the swell-consolidation method and the swell-reload method is that the latter applies compression to the specimen in steps, with an interval of 24 h between each step [9], which ensures the full swelling of the specimen and also increases related errors such as friction; the swelling pressure is the vertical stress required to restore the specimen to its original state. The "extra" macroscopic void fraction generated during the initial swelling at low pressure results in higher swelling pressure

measured by the swell-consolidation method compared to the constant volume method [21]. The double consolidation test was performed on two samples, one of which was wetted, swollen and completely submerged, and its maximum expansion was recorded after 24 h of expansion, while the other was compressed at natural moisture content, and the amount of compression was equal to the swelling of the wetted sample; the vertical stress was defined as the swelling pressure [14]. Apparently, the constant load method is constrained by the free-swelling method [14]. The values measured by the double consolidation test are significantly higher than the values obtained by the free swelling test [22].

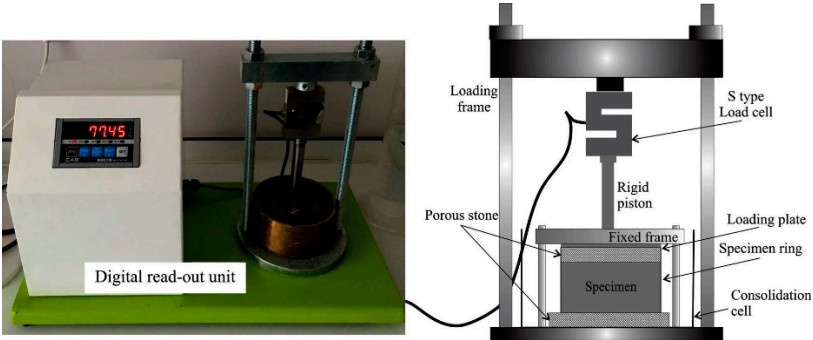

**Figure 1.** Schematic view of the experimental setup using a constant-volume cell, according to [14].

The PS-test is a simple, reasonable, and reliable swelling test based on the improvement of the loading of the tested specimen; its validity has also been confirmed by other numerous tests. Its most important advantages are that it does not require a complicated setup or design, uses conventional consolidation equipment, and does not need the provision of constant volume conditions. On the other hand, the disadvantage is that two samples are needed at the same time. The principle is that the response of one specimen is used to refine the load step of the second specimen and that a small incremental load-controlled swelling does not cause excessive water absorption in the soil specimens [14].

Many optimization test schemes were proposed according to the different properties of the above-mentioned test methods. For example, as early as in 1975, Bishop et al. [23] proposed a triaxial instrument that could directly measure the vertical swelling of a sample in a hydraulic triaxial stress path cell; it could predict the swelling potential of the soil under test better than the consolidation test method [11]. A new constant-volume sensor, developed by Tang et al. [9], could measure the swelling pressure without both any strain adjustment and the influence of test equipment stiffness, thereby reducing the measurement errors (Figure 2). To reduce diaphragm deformation under external pressure, a mercury-filled pore gap is introduced below the diaphragm in the BER-A-58S pressure sensor utilized in the upper portion. The upper-pressure transducer is in contact with the upper surface of the sample for the direct measurement of the swelling pressure. Figure 3 shows the diagram equipped with this pressure transducer used with the constant volume method without manual load adjustment, which makes it very convenient for monitoring the long-term changes in swelling pressure.

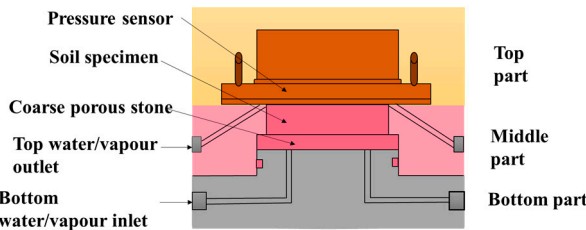

**Figure 2.** Schematic view of the developed constant-volume cell [9].

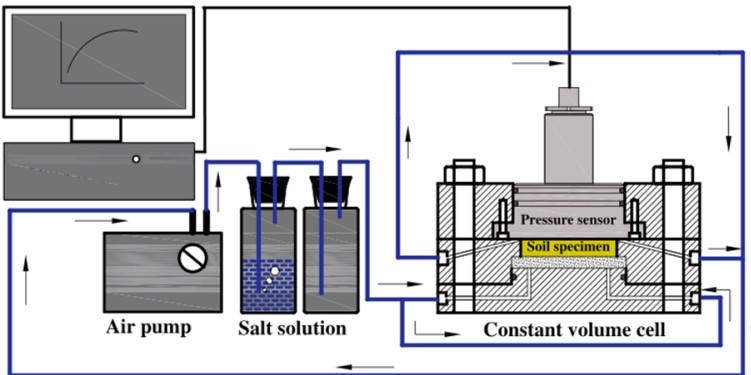

**Figure 3.** Schematic view of the experimental setup using the constant-volume cell [9].

Accuracy, directness, simplicity, and long-term monitoring have become important criteria for selecting a test method. In addition, the different loading and wetting conditions used by a method will also cause the achieved swelling pressure to behave differently [11]. For instance, at constant volume, the swelling pressure is surprisingly reduced after wetting in soil subjected to a high initial vertical stress [9].

In recent years, microstructural investigations of the bentonite swelling pressure have clarified the causes of swelling pressure variation in relation to the test method. It was shown that similar swelling pressures obtained by different tests depended on their similar micro- or macro-porosity. The swelling pressure will show a good consistency, independent of the test method, when the samples are wetted under a sufficiently high vertical stress [21]. In general, the constant volume test is still the most commonly used method in current research [24–26].

## 3. Influence of Thermo-Chemical Effect on the Swelling Pressure

This review mainly discusses the changes of the swelling pressure from the perspective of crystal layer swelling and double electric layer swelling in the presence of salt solutions and in different temperature conditions, almost ignoring the influence of mineral alteration during this process.

Generally, the crystalline-layer swelling mechanism involving the internal minerals and the diffusive double-layer swelling mechanism involving the mineral particles are considered to be the main swelling mechanisms of bentonite [27] and are commonly considered to analyze the changes in swelling pressure in complex environments [28]. As shown in Figure 4, the wedging of water molecules into the crystalline layers leads to the crystalline layer swelling. The swelling changes from crystalline-layer swelling to double-electric-layer swelling when the number of water layers in the crystalline layers reaches a certain value [27,29,30]. Both swelling mechanisms are referred to as interlayer swelling. Interlayer swelling above 22 Å is associated with double-layer swelling, and interlayer swelling between 10 and 22 Å is associated with crystalline layer swelling caused by the adsorption of water molecules and is determined by the layer charge of clay minerals, exchange cations, and the attraction between water molecules and polar surface groups [31–34]. The crystalline layer interval and the thickness of the diffusion double layer are essential factors in the study of the magnitude of the swelling pressure, and changes in temperature and pore solution can lead to changes in both, thus affecting the magnitude of the swelling pressure.

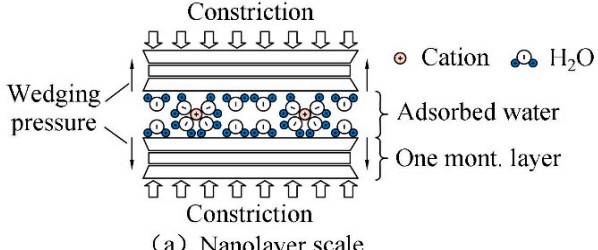

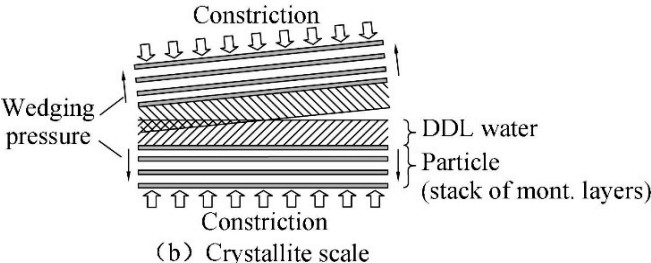

**Figure 4.** Formation of the wedging pressure at (**a**) a nanolayer scale and (**b**) a crystallite scale [35].

*3.1. Effect of the Pore Salt Solution Concentration on the Swelling Pressure*

Studies have shown that the increase of the pore solution concentration leads to a decrease in bentonite expansibility [3,36,37]. Different types of solutions have different effects on the swelling pressure of bentonite.

The concrete material in an HLW repository is gradually degraded during the interaction with pore water producing an alkaline solution, while montmorillonite, the main mineral component of bentonite, is dissolved gradually by the alkaline solution [38,39].

As shown in Figure 5, GMZ01 bentonite with a montmorillonite content up to 75.4% was tested for its constant-volume swelling pressure in the presence of different saline solution concentrations and types of infiltration, while maintaining the temperature at $20 \pm 1$ °C [26]. According to the classical double- layer theory, increasing the solution concentration will inhibit the swelling of the double layer [40], and the thickness of the diffusion double layer is inversely proportional to the square root of $Na^+$ concentration [16]. Therefore, high-concentration saline solutions will reduce the repulsive force between the particles, leading to a decrease in the swelling pressure. It is obvious that the ion types, $Ca^{2+}$ and $Na^+$, had similar effects on the swelling pressure at low concentrations, but when the concentrations were greater than 0.5 M, the inhibition of the swelling pressure of bentonite by $Ca^{2+}$ was not so obvious compared to that induced by $Na^+$. A greater base spacing of Ca-based montmorillonite under constant volume conditions was reported [41], and Chen et al. [42] suggested that $Ca^{2+}$ replacement changed the pore fabric characteristics, leading to a greater crystalline-layer swelling deformation. However, with regard to the diffusive double-layer swelling, the Gouy-Chapman model [43,44] suggested that the higher the electrolyte valence type and the electrolyte concentration, the smaller the double-layer thickness.

Several researchers attribute the greater swelling pressure in the presence of $Ca^{2+}$ to the fact that its action produces larger and thicker particles [45], and the bilayer effect is not so pronounced; so, bentonite has greater mechanical strength and greater swelling pressure.

As shown in Figure 6, where the effect of temperature was ignored, the constant-volume swelling pressure test was performed on GMZ Na–bentonite with 75.4% bentonite, initial water content of 11.92%, dry density of 1.7 Mg/m$^3$; the swelling pressure curve showed three phases [24]. Analogously, when the dry density was high, 1.7 Mg/m$^3$, and the initial water content was relatively low, i.e., 10.76%, the swelling process under the action of the saline solution comprised three stages: firstly, crystalline swelling, followed by the breakup of quasicrystals, and then double-layer swelling [26]. The first stage was characterized by a rapid increase in the swelling pressure, marked by increased interlayer

cation hydration, and as the cation concentration continued to increase, water molecules within the crystals left the crystal layer, which gradually inhibited the swelling of the crystal layer [46]. The second stage was the transition from crystalline to permeable swelling, with a small decrease in the swelling pressure due to the swelling of soil particle agglomerates and the collapse of the soil skeleton, which was manifested by the splitting of thick quasicrystals into fine quasicrystals and the filling of large pores (inter-agglomerate pores). The third stage was the double-layer swelling stage, where the swelling pressure increased again, and the crystalline swelling became less pronounced; for specimens infiltrated with the salt solution or distilled water, the final swelling pressure in this stage was maintained in equilibrium [47].

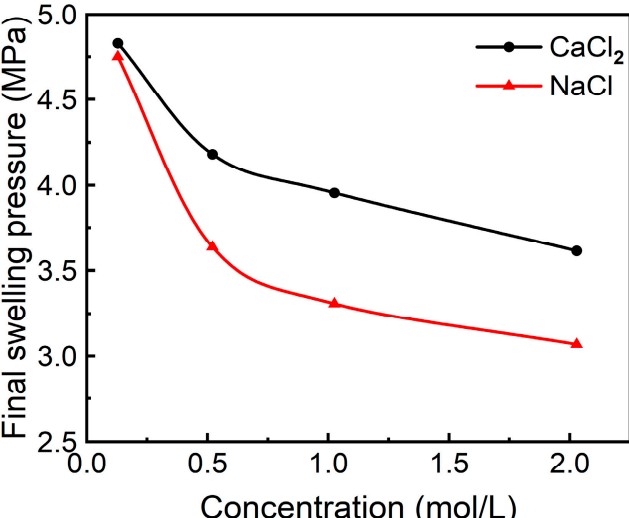

**Figure 5.** Comparison of the influence of the cation type on the swelling pressure of GMZ01 bentonite [26].

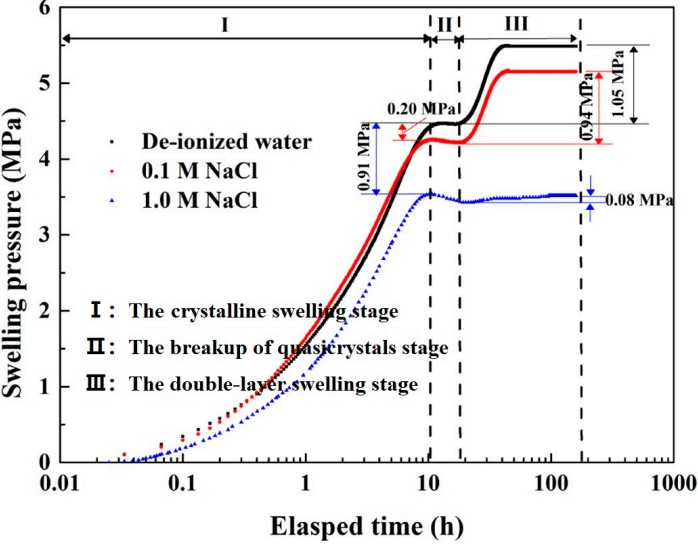

**Figure 6.** Effect of a salt solution at different concentration on the swelling pressure [24].

Zhang et al. [48] proposed to distinguish the two swelling mechanisms by the pore ratio. Similarly, ignoring the temperature change, the swelling pressure of GMZ01 bentonite with 75% montmorillonite content was investigated, and the combined effect of double-layer swelling and crystal-layer swelling was distinguished from the swelling caused by the crystal-layer swelling effect with a critical pore ratio of 0.41. In the case of low porosity (less than 0.41), a diffusion double layer could not form [49], and the swelling pressure

increased with the increasing concentration but was not particularly pronounced. Thus, the swelling pressure could be explained by the theory of crystalline-layer swelling with increasing cation valence, on the basis of which, $Ca^{2+}$ will lead to a greater crystalline-layer swelling compared to $Na^+$ [42]. The swelling pressure increased with the increasing pore solution concentration for pore ratios less than 0.5 [50]. Combined with the above analysis, this stage was considered to be the crystal-layer swelling control stage, and the swelling pressure was almost generated by the swelling between the crystal layers.

The above study investigated the variation of the swelling pressure from the perspective of the pore ratio under the action of two swelling mechanisms, while some studies further analyzed the reasons for the variation of the swelling pressure from the perspective of the variation of soil–water interfacial forces. For illustration purposes, the concepts of accumulated wedging pressure (AWP) and dissipated wedging pressure (DWP) were used to determine the development of the swelling pressure in terms of the competition between the two pressures [35].

The wedging pressure results from the wedging of water molecules into the interlayer of montmorillonite, interparticle swelling, and the competition between the dissociation of the initially accumulated montmorillonite layer and aggregated particles, due to the continuous entry of water molecules and the deformation, damage, and dislocation of the bentonite block, leading to pressure dissipation. It manifests as small pore formation and large pore collapse when the latter process predominate [51]. The concept also links the flatness of the pore structure to the ultimate stabilization of the swelling pressure, the essence of which remains inseparable from the swelling mechanism of the mineral crystal layer and the interparticle double layer.

### 3.2. Effect of Temperature on the Swelling Pressure

Elevated temperatures can have an impact on bentonite and its engineering properties. For example, simulations of high-discharge waste disposal in China have shown that thermal swelling and saturation processes may affect a tank stability for long periods [52], as higher temperatures can provoke montmorillonite dehydration [53]. The effect of temperature on the swelling pressure is highly controversial. Liu et al. [25] used distilled water and performed constant-volume swelling pressure tests on GMZ bentonite particle mixtures at different temperatures. It was shown that the swelling pressure changed faster and gradually increased with higher temperatures (Figure 7). A similar phenomenon was also indicated by some related studies [5,54]. Nevertheless, other tests also showed that the swelling pressure decreased with the increasing temperature [55–58].

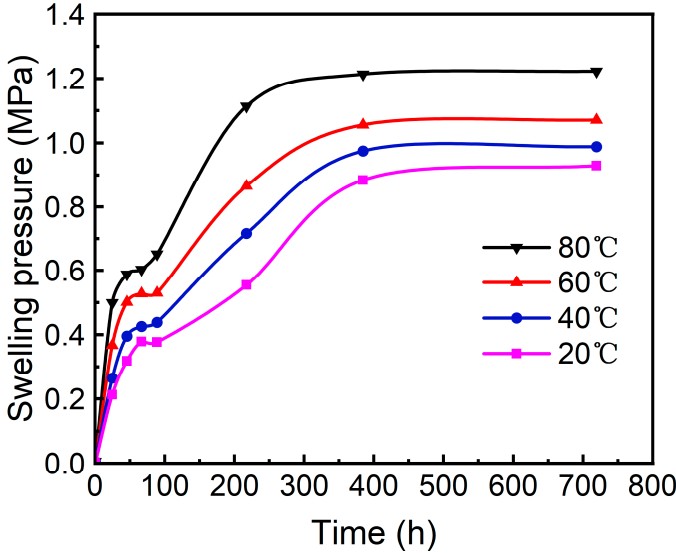

**Figure 7.** Evolution of the swelling pressure of a bentonite pellet mixture at different temperatures [25].

In contrast, some studies have shown opposite effects of temperature on different bentonite types, such as heating leading to an increase in the swelling pressure of Na-based bentonite and a decrease in the swelling pressure of Ca-based bentonite [59]. This can be attributed to two main mechanisms: osmotic pressure and the net effect of crystalline lattice shrinkage. Lattice shrinkage caused by the dehydration of the interlayer space and the increase in osmotic pressure at the interlayer contact is different for the two clays, with lattice shrinkage dominating in Ca-based bentonite, and the increase of osmotic pressure dominating in Na-based bentonite. Different bentonite types produce different responses to temperature variation; therefore, combined with the swelling mechanism, it can be concluded that the effect of temperature on hydration greatly influences the variation of the final swelling pressure of bentonite.

Hydration causes an increase in the interlayer pore volume of clay minerals, leading to the swelling of clay particles and, thus, to the formation of clay aggregates [60]. The bentonite swelling potential is influenced by the water content due to the coupling of micro- and macro-structures caused by hydration [61]. As shown in Figure 7, a test was conducted by injecting distilled water at the bottom of the specimen, and the results were analyzed. Firstly, the water entering from the bottom caused the soil particles to swell and densify, and thus the swelling pressure increased rapidly; however, the swollen particles prevented further water penetration, which slowed down the increase in the swelling pressure. Finally, the reduced strength of the wetted particles was counteracted by the rearrangement of the upper particles, and thus a plateau in the development of swelling pressure occurred [25]. The dual-structure model may explain the reduction of the particle strength in the wetted state [62,63]: on the one hand, wetting increased the interlayer distance between the clay flakes leading to the swelling of the clay aggregates, and on the other hand, wetting weakened the macro-structural resistance.

Liu et al. [25] provided a comprehensive explanation of the relationship between swelling pressure and temperature, indicating that the variation of the swelling pressure with temperature depends on the competition between hydration pressure, hydraulic pressure, and osmotic pressure thermo-tropic response. The Na-based GMZ bentonite with a dry density of 1.45 mg/m$^3$ in the study exhibited an enhanced swelling pressure as it possesses a large number of pores between the particles, and the temperature-dependent increase of its swelling pressure was affected by the enhanced water pressure and osmotic pressure masking the limited decrease in hydration force. Therefore, the effect of temperature on the swelling pressure depends not only on hydration but also on the pore water pressure and electrolyte properties.

*3.3. Coupled Thermo-Chemical Effect on the Swelling Pressure*

Chen et al. [28] conducted alternating wetting tests using distilled water and NaCl solutions of different concentrations on Na–GMZ bentonite with an initial dry density of 1.7 g/cm$^3$ at 20 °C and 60 °C, as shown in Figure 8. The results showed that when the thermo-chemical effect was coupled, the increase of temperature and electrolyte concentration was not conducive to the development of swelling pressure, and in the multi-step salination–desalination process, the initial swelling pressure of the sample could not be fully restored after saline circulation. The possible causes were the irreversible interpore collapse and the further homogenization of the clay structure at the macrostructural level.

Similar results were obtained with the swelling pressure test of monovalent bentonite with the same dry density, in India [58]. In addition, that study showed that a high temperature can promote the expansion of bivalent bentonite. In general, the swelling pressure of the divalent bentonite increased monotonically and then stabilized, while the collapse occurred in the middle of the monovalent bentonite and increased again after equilibrium. When the thermo-chemical effect was coupled, the swelling pressure of monovalent bentonite was greatly reduced, but that of divalent bentonite was very small. Compared with the equilibrium swelling pressure when using distilled water wetting at 25 °C, the inhibition effect of a 0.1 M NaCl solution was less than the promotion effect of

95 °C temperature on the swelling pressure. Therefore, under the condition of 95 °C and 0.1 M NaCl solution infiltration, the equilibrium swelling pressure of bivalent bentonite still increased. However, when the sample was soaked in a $CaCl_2$ solution with the same concentration and at the same temperature, the equilibrium swelling pressure of bivalent bentonite was reduced. The authors attributed this to the combined effect of high temperature, leading to the expansion of the soil particles, and the increase of microscopic pore water pressure. At the same time, when low-concentration $Na^+$ was exchanged with $Ca^{2+}$, the exchanged $Na^+$ was hydrated until the exchange process was saturated, which promoted permeability and expansion [46]. However, this study did not take into account the phenomenon of mineral alteration occurring in bentonite. The microscopic mechanism of bentonite swelling pressure variation under coupled thermo-chemical effects needs to be further refined and studied.

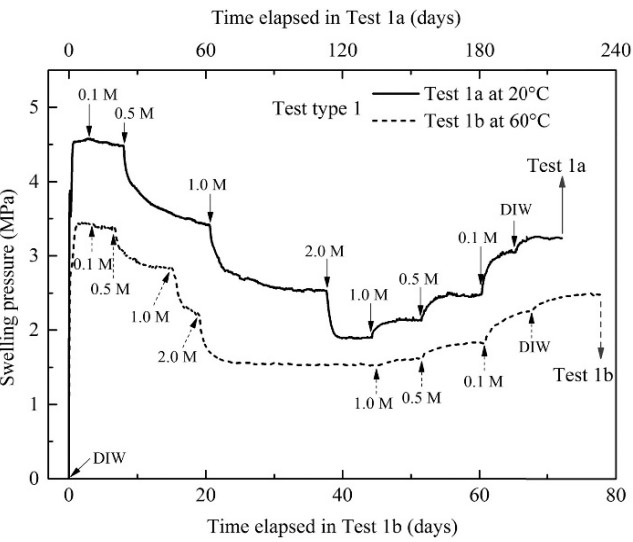

**Figure 8.** Evolution of the swelling pressure under thermo-chemical effects [28].

## 4. Effect Mechanism of Swelling Properties

Generally, the mechanistic explanation of macroscopic phenomena ultimately requires microscopic-scale evaluations. Thermo-chemical effects will have a great impact on the microstructure of the soil, and as a consequence, it is essential to consider the variations occurring in the microstructure of bentonite with the hydration process when describing the expansion behavior of bentonite. Aside from pore distribution, another important factor affecting the microstructure of soil is water distribution in the pores, which has a strong control effect on the mechanical and seepage characteristics of unsaturated soil.

### 4.1. Pore Distribution

Clay represents a complex porous structural body, with clay layers and interlayer pores composing a basic unit of clay particles, such as montmorillonite lamellae and interlayer pores, as shown in Figure 9 [60]. Water in clay soils includes microstructural water and macro-structural water [64]. As shown in Figure 9, water in interparticle and interlayer pores is microstructural water, i.e., adsorbed water, and water in interaggregate pores is macrostructural water. The surface of the clay particle unit carries a negative charge, and a diffuse double layer is formed between the clay particles, such as the diffused double-electric-layer water layer at the crystallite scale, shown in Figure 4b. The clay particles and the inter-clay pores form the basic clay aggregate, then the aggregates and the inter-aggregate pores develop a representative basic clay unit. Consequently, the clay block can be roughly divided into interlayer pores, intergranular pores, and inter-aggregate pores at three levels of pore space, and fractures visible to the naked eye develop from the inter-aggregate pores.

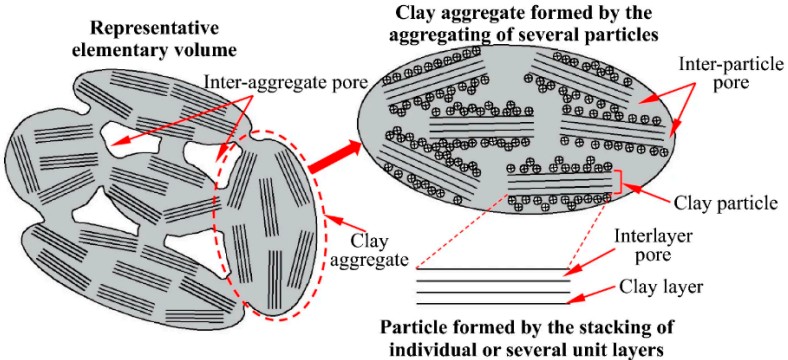

**Figure 9.** Fabric units and pore spaces of compacted clay (modified after [2,65]).

Investigating bentonite microstructure entails the study of the pore structure of bentonite as well as its mineral composition. Freeze–drying techniques [64,66] and mercury intrusion pore size measurement (MIP technique) [67,68] were used to obtain PSD distribution curves and thus distinguish between macro- and micropores. Nevertheless, the limitations of the MIP technique itself [69–71], combined with the differences in clay types, led to different pore size measures. For example, when the specimen was a mixture of bentonite and Callovo–Oxfordian (COx) claystone with low dry density, the average pore sizes of its three main pore groups were 10–22 μm for macro-pores, 0.17–0.52 μm for mesopores, and about 25 nm for small pores [60].

Ma et al. [72] investigated the microstructure of saturated bentonite samples with a moisture content of 0.46 in the wet–dry cycle and found that micro-pores and macro-pores corresponded to 12 nm and 6 μm; they considered the size of 500 nm as a threshold to distinguish between intra-aggregate and inter-aggregate pores. Zhang et al. [21] performed MIP analysis of compacted MX80 Wyoming bentonite samples with 80–92% montmorillonite content and identified small and large pore groups with major pore sizes of 17 nm and 21 μm, respectively; pores with a diameter of less than 100 nm were considered micro-pores [35]. In addition, scanning electron microscopy (SEM) tests can provide high-precision clay images to study the structural changes of clays more visually, and X-ray diffraction (XRD) can evaluate the changes in the composition of clay minerals to determine the causes of changes in swelling pressure [24,73], as well as measure variations in mineral layer spacing as a support condition for changes in the swelling pressure [74]. The structure determines the properties, and the complex structure of clays dictates that variations in swelling properties are significantly influenced by changes in pore distribution and pore water status.

*4.2. Pore Water Status*

Pore water refers to water that is confined to soil pores or clay minerals, and the properties of pore water are changed by the interaction between soil particles and pore water. The pore water status of soils also affects their mechanical properties to some extent. Studies on soil moisture characteristics suggested that soil moisture is controlled by two mechanisms: capillary and adsorption [75,76]. Adsorption predominates under high suction, and capillary dominates under low suction [77,78]. Ma et al. [72] tested the wet–dry cycle with different suction values in powder and compacted Ningming expansive soils. When the suction reached 24 MPa, the SWCCs curves of the two kinds of soil samples coincided. Therefore, the suction pressure of 24 MPa was defined as the threshold for capillary and adsorption of the soil samples, distinguishing capillary and adsorption effects in expansive soil with a quantitative suction magnitude. Meanwhile, some constant-volume experiments provided similar results [79]. The variation of absorbed water content in the soil with temperature is controlled by two mechanisms: pore water potential and soil structure. The pore water potential of adsorbed water is smaller than that of capillary water and decreases with the decrease in temperature [80]. According to Le Chatelier's

principle [81], an increase in temperature facilitates the desorption process and inhibits the exothermic process. According to the thermodynamic theory Equation (1), both cation and surface hydration processes are exothermic because the adsorbed water is adsorbed on the crystal surface, the entropy of the system decreases, $\Delta S_a < 0$, and then the enthalpy of adsorption becomes $\Delta H_a < 0$; therefore, adsorption is exothermic [82]. The decrease in temperature enhances hydration, influencing the entropy of the hydration reaction [83].

$$\Delta G_a \;=\; \Delta H_a - T\Delta S_a \;=\; 0 \tag{1}$$

The absorbed water content increases with the decrease in temperature, which can be attributed to the decrease in pore water potential and the increase in micro-pore volume, as shown in Figure 10. Depending on the Yang–Laplace equation, capillary suction can be described by Equation (2) [82]. As the temperature increases, under a given suction Pc, the water-saturated pore diameter d decreases, with the decrease of surface tension (σ) and contact angle (θ) the faster the speed decreases. This results in a reduction in the amount of capillary water [84,85]. Thus, the conversion of adsorbed water to capillary water with inreased temperature and the further loss of capillary water will be detrimental to the retention of water in clays. Liu et al. [82] demonstrated that the water retention capacity of three bentonite samples decreased with increasing temperature for a given suction condition. In both restricted- and free-swelling conditions, bentonite water retention capacity will decrease with the increasing temperature (20–80 °C), and this is especially evident when the suction is low [57].

$$P_c \;=\; \frac{4\sigma\cos\theta}{d} \tag{2}$$

The adsorbed water content is simultaneously controlled by the pore solution concentration, which increases to the detriment of the adsorbed water [80]. It is clear that an increase in the pore solution concentration will inhibit the development of the bilayer thickness, according to Equation (3). However, the role of salt concentration on crystal layer swelling needs to be further investigated. As regards the changes in pore structure, the adsorbed water content is affected by the salt concentration in the same way as the micro-pore volume. Norrish et al. [86] concluded that, although salt concentrations less than 1 M have little effect on the swelling of crystalline layers, the increased electrolyte concentration leads to the dehydration of montmorillonite by osmosis, which is detrimental to the occurrence of inter-montmorillonite hydration. Thus, high salt concentrations have an inhibitory effect on hydration.

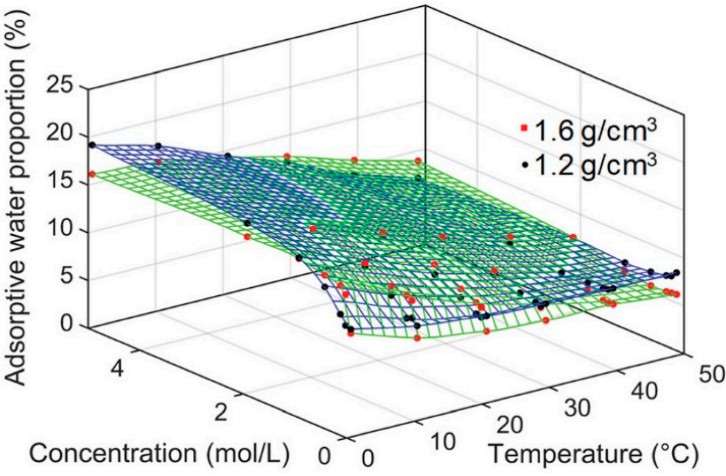

**Figure 10.** Variations of the adsorptive water content of samples at different dry densities [80].

The thickness of the double electric layer is calculated by [40]:

$$\frac{1}{K} = \left( \frac{\varepsilon^0 DkT}{2n^0 e^2 v^2} \right)^{\frac{1}{2}} \tag{3}$$

$\frac{1}{K}$ is the thickness of the double layer, $\varepsilon^0$ is the vacuum coefficient, $D$ is the dielectric constant, $k$ is the Boltzmann constant, $T$ is the temperature, $n^0$ is the pore solution concentration, $e$ is the unit charge, and $v$ is the ionic valence.

Significantly, because of the thermo-chemical effects on pore water, as shown in Figure 10 [80], the volume of micro-pores decreased, and the volume of macro-pores increased with increasing salt concentrations, and micro-pores were more frequent at low temperatures. In addition, it was observed that more water was adsorbed at low temperatures and in high-salt-concentration environments, which seems to be contrary to the theory. By investigating the effective activation energy [87], which is known as the energy barrier for water molecule conversion, Tian et al. [87] found that elevated salt concentrations accelerated the water molecule relaxation and produced a higher effective activation energy compared to samples saturated with distilled water, enhancing water molecule adsorption to some extent. In reality, such water with a short relaxation time is somewhat different from adsorbed water or exists in the form of hydrated ions surrounded by water molecules, an effect that is more pronounced in the presence of saline solutions, masking the inhibition of the double electric layer and the reduction of the micro-pore volume by the increase of the pore solution concentration; hence, the higher measured adsorbed water content [80]. Bentonite exposed to a concrete-leaching groundwater solution introduced at elevated temperatures showed a decrease in macro-pore volume and an increase in micro-pores, with the average pore size decreasing from 6 nm to 4 nm [88]. This phenomenon suggests that a high-temperature situation may also lead to an increase in micro-pores; so, the study of adsorbed water content should take into account the pore structure in the case of coupled thermo-chemical effects.

In conclusion, dry and wet cycles destroy the soil pore structure and weaken the water retention capacity of bentonite; elevated temperatures reduce the water retention performance of bentonite; increased salt concentrations are detrimental to the development of micro-pores and the adsorption of adsorbed water, thereby reducing the swelling capacity of bentonite. When the thermo-chemical effects act simultaneously, combining with changes in soil structure, thermodynamic properties, and ion hydration effects, their influence on the pore water status becomes more complex. As a consequence, more in-depth studies are essential.

*4.3. Intergranular Stress*

At present, studies on the swelling mechanism of bentonite generally remain at the microscopic level. A few scholars have proposed a new formula to determine the effective stress in unsaturated soil from a macroscopic perspective, which can describe the stress state in unsaturated soil well [89,90]. In 2014, Wei [91] proposed a theory of intergranular stress acting between soil particles, which comprehensively characterized the physicochemical interactions between soil particles from a macroscopic perspective.

Wei [91] introduced the concept of equilibrium solution and pointed out that the measured pore pressure is not the same as the true pore pressure within the soil. For bentonite, when the soil is saturated with an aqueous solution, the magnitude of the water pressure within the pore space is not uniform due to the physicochemical interaction between pore solution and soil water. The equilibrium solution is defined as the water in a measurement vessel that has reached equilibrium with a pore solution without surface action of the soil particles. The soil is in equilibrium with the sensor, when measuring the pore pressure. In other words, we determine the pore pressure by measuring the pressure of the equilibrium solution, whereas the instrument measures only the potential energy of the water and not the mechanical pressure.

Wei [91] deduced the expression of true pore pressure considering that the water potential energy in the pore is the same as that in the container at an equilibrium state:

$$p^l = p^l_w + \Pi \tag{4}$$

where $p^l_w$ is the actual pore water pressure in the soil, and there is a difference $\Pi$ from the measured pore pressure $p^l$; $\Pi$ is called generalized osmotic pressure. Intergranular stress is defined as:

$$\sigma'' = \sigma - (p^l_w + \Pi) \tag{5}$$

where $\sigma''$ is the intergranular stress; $\sigma$ is the total stress.

This generalized osmotic pressure reflects the interaction between soil particles and pore water and is divided into two components: the Donnan osmotic pressure and the pressure derived from surface forces. The Donnan osmotic pressure relates to the concentration difference between the pore solution and the equilibrium solution. The second component of generalized osmotic pressure is the pressure caused by surface forces, including the surface tension of the interface, the electrostatic force, Van der Waals gravity, electric double-layer repulsion, etc. The equation is:

$$\Pi = \Pi_D - \rho_\oplus^{l_{H_2O}} \Omega^l \tag{6}$$

In the equation, $\rho_\oplus^{l_{H_2O}}$ is the mass density of pure water, equal to 1.0 g/cm$^3$, $\Pi_D$ is the Donnan osmotic pressure, and $\Omega^l$ is the surface potential energy caused by microscopic surface forces. Donnan osmotic pressure is the pressure caused by the pressure difference betwe en the internal and the external concentration of the soil related to the porosity and solution concentration in the soil. It is expressed by:

$$\Pi_D = \frac{RT\rho_\oplus^{l_{H_2O}}}{M_{H_2O}} \ln\left(\frac{a_A^{l_{H_2O}}}{a^{l_{H_2O}}}\right) \tag{7}$$

where $M_{H_2O}$ is the molar mass of water, equal to 18 g/mol, $a_A^{l_{H_2O}}$ is the activity of water in the equilibrium solution, which is equal to the molar fraction of water in the equilibrium solution in the ideal solution, $a^{l_{H_2O}}$ is the activity of water in the soil pore solution, which is equal to the molar fraction of water in the soil pore solution in the ideal solution. $\Omega^l$ is expressed as:

$$n^l\rho^l\Omega^l(T,n^l) = n_0^l\rho^l\Omega^l(T,n_0^l) + \int_{n^l}^{n_0^l}\left[s_M(T,n^l) - \Pi_D(T,n^l)\right]dn^l \tag{8}$$

where $n^l$ is the volume fraction of water, defined as the volume of water in the pore as a proportion of the total volume, $p^l$ is the density of water, for dilute solutions, $\rho^l \approx \rho_\oplus^{l_{H_2O}}$, $s_M$ is the measured (or controlled) matrix suction, $p^l_W$ is the measured pore pressure, $n_0^l$ is the volume fraction of pore water at saturation, equal to the porosity $n$.

The intergranular stress expression can consider the physicochemical effects between particles and unitize the infiltration, capillary, and adsorption effects caused by physicochemical interactions, which can better describe the behavior of bentonites in terms of strength and deformation. Moreover, the capillary and physicochemical pressure at the microscopic scale can be better applied to the intergranular stress expression at the macroscopic scale, in unsaturated soils.

Ma et al. [92] experimentally researched the effect of pore solution on the swelling pressure of expansive soils, pointed out that the variation of the swelling pressure is determined by the intergranular stress, described the swelling pressure variation from the perspective of macro-mechanics, and proposed a model of swelling force prediction. The superiority of this model for swelling pressure prediction was confirmed by validating

it on several bentonites with different properties (Figure 11). Overall, studies of the effect of intergranular stress on the swelling pressure of bentonite under thermo-chemical effects are still relatively rare and lack extensive experimental validation; therefore, further investigation is necessary.

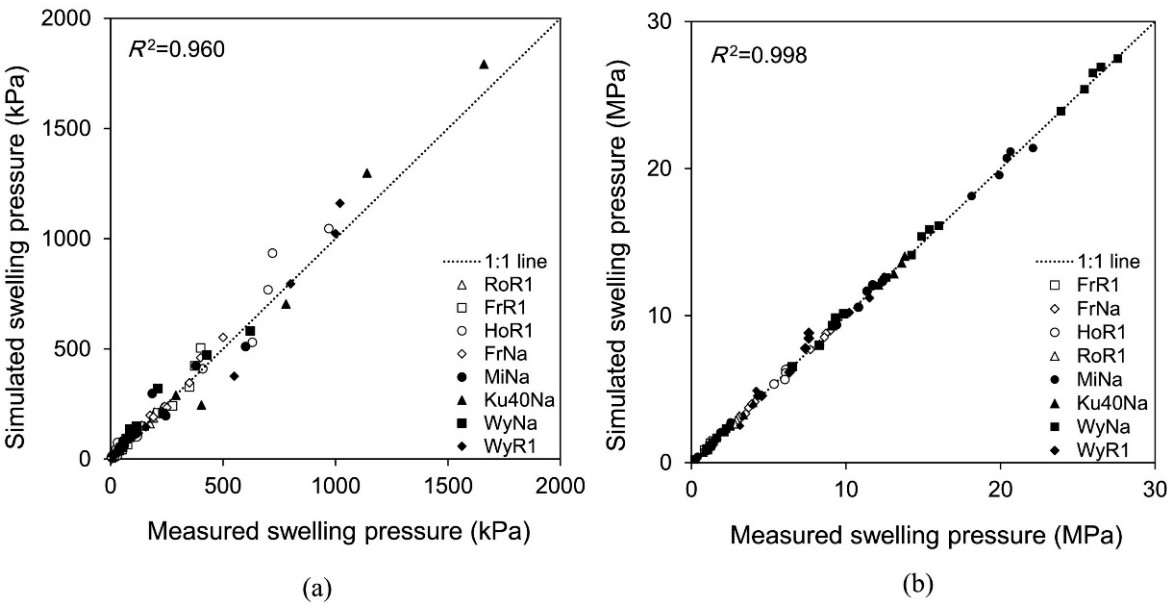

**Figure 11.** Comparison of measured and predicted swelling pressures for eight soils [93]: (**a**) low swelling pressure; (**b**) high swelling pressure [92].

### 5. Conclusions and Prospects

In recent years, scholars have conducted elaborate studies on the swelling pressure of bentonite in the deep geological repositories of HLW, focusing on experimental methods, influencing factors, change patterns, and intrinsic influencing mechanisms. However, considering the coupled thermochemistry and hydrodynamics phenomena involved in the actual nuclear waste containment process, the existing research findings still have certain limitations. Additional research is desired in the future in the following areas:

(1) Various test methods will result in different values of measured swelling pressure. A comparative analysis of different test methods should be conducted, specifically considering the coupled environmental effects in the nuclear waste containment process to identify the optimal test method.

(2) Diffuse nuclear radiation heat is a major influencing factor in nuclear waste containment. Considering the effect of temperature, studies on the hydrodynamic properties of bentonite remain to be conducted.

(3) Analysis of the mechanism of swelling pressure changes: from the microscopic perspective, the process and mechanism of swelling pressure changes still lack profound recognition. In addition, from the macro-mechanics perspective, there are only sporadic studies which generally lack experimental verifications. In the future, the analysis of the swelling pressure variation law and the internal mechanism of swelling pressure evolution needs to be further refined.

**Author Contributions:** Conceptualization, J.L. and C.Y.; validation, J.L., W.S. and Y.Z.; investigation, J.L., W.S. and Y.Z.; resources, J.L., W.S. and Y.Z.; writing—original draft preparation, J.L.; writing—review and editing, J.L. and C.Y.; supervision, C.Y.; funding acquisition, C.Y. All authors have read and agreed to the published version of the manuscript.

**Funding:** This research was funded by National Natural Science Foundation of China (Grant No. 52109133), Shanghai Sailing Program (Grant No. 21YF1432700).

**Institutional Review Board Statement:** The study did not involve humans or animals.

**Informed Consent Statement:** The study did not involve humans or animals.

**Data Availability Statement:** All data, models, or codes that support the findings of this study are available from the corresponding author upon reasonable request.

**Acknowledgments:** This research was supported by the National Natural Science Foundation of China (Grant No. 52109133), Shanghai Sailing Program (Grant No. 21YF1432700).

**Conflicts of Interest:** The authors declare no conflict of interest.

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
