# Peer review of "Research Progress on the Influence of Thermo-Chemical Effects on the Swelling Pressure of Bentonite"

_applsci, doi:10.3390/app13095580_

Round 1
Reviewer 1 Report
1. The authors quote drawings from other publications. Probably it would be necessary to put a slightly different signature under the drawings. For example: Fig. 1. The scheme of the experimental setup using a constant volume cell according to [13]. 2. A few minor flaws: Line 381 represents equation (2) Line 492 - [94 ]
Reviewer 2 Report
General comments: This paper systematically describes the progress and limitations of the researches on bentonite swelling pressure under the effect of thermochemistry, and the outlook on the future research and development direction of bentonite swelling pressure are provided. In general, this article is well written in English, but some sentences and grammar could be improved. The authors do not pay attention to some small details and additional explanations are needed.
Comment 1: Lines 100-106, “the free swelling method is constrained by the constant load method”, the relationship between the double consolidation test and free swelling method is not clarified, please specify whether there is any error here.
· Comment 2: Lines 138-139, “the swelling pressure is surprisingly reduced after wetting, for soil subjected to a high initial vertical stress”, “high initial vertical stress”, in what conditions, exactly?
· Comment 3: Lines 350-354, what is the basis for defining 24MPa as the threshold between capillary and adsorption?
Comment 4: Lines 420-421, “increased concentrations are detrimental to the development of micro-pores and thereby reduce the swelling capacity”. According to the content of the discussion, it seems that the reasons for the reduction of the expansion capacity are not comprehensive enough.
· Comment 5: In section 4.3 intergranular stress, the authors point out that “a few scholars have proposed to explain it in terms of macroscopic mechanical parameters”. However, only the macroscopic explanation of intergranular stress is mentioned in the paper. Please supplement the researches on macroscopic mechanical.

Reviewer 3 Report
The manuscript reviewed the thermo-chemical effects on the swelling pressure of compacted bentonite. The topic is important in radioactive waste disposal; however, the manuscript includes technical items for which reconsideration and major revision are required.
(1) Overall, it was unclear what is technical issue in radioactive waste disposal relative to recent experimental and analytical progresses. The perspective regarding thermal and chemical are coupled or not should be clarified in the manuscript. Destination of further research would be informative for the reader.
(2) Section 2: Technical term "swelling pressure" should be obviously defined. It can be ideally defined, for instance, the pressure to restrain the volume of compacted expansive clay when it absorbs solution. It is confused to understand the difference of consolidation pressure and swelling pressure for the reloading test. In addition, the swelling pressure is affected by restrained condition. The following paper may be helpful in understanding the different results in several types of swelling test.
Géotechnique 73(67):1-35. https://doi.org/10.1680/jgeot.20.P.348
(3) Section 3: The literature reviewed seems to be biased. For example, regarding the evolution of swelling pressure, the mechanism described on pages 195-209 does not explain the swelling pressure of bentonite of the same dry density with different initial water content. For an objective and comprehensive mechanism, the following paper may be of interest.
Géotechnique. https://doi.org/10.1680/jgeot.21.00312
(4) The scope of your review should be clarified. It should be stated that chemical alteration (mineralogical alteration) is not covered. Also, the distinction between what is problematic in coupled and what is problematic in thermal and chemical, respectively, was also unclear.
(5) Many editorial errors, including the number of figures and equations, were made on the manuscript.
Round 2
Reviewer 3 Report
The manuscript was revised along the reviewer's comment. It can be published in the present form.